

# Quantifying non-Markovianity in open quantum dynamics

**Chu Guo**

Henan Key Laboratory of Quantum Information and Cryptography,
Zhengzhou, Henan 450000, China
Key Laboratory of Low-Dimensional Quantum Structures and Quantum Control
of Ministry of Education, Department of Physics and
Synergetic Innovation Center for Quantum Effects and Applications,
Hunan Normal University, Changsha 410081, China

guochu604b@gmail.com

## Abstract

Characterization of non-Markovian open quantum dynamics is both of theoretical and practical relevance. In a seminal work [Phys. Rev. Lett. 120, 040405 (2018)], a necessary and sufficient quantum Markov condition is proposed, with a clear operational interpretation and correspondence with the classical limit. Here we propose two non-Markovianity measures for general open quantum dynamics, which are fully reconciled with the Markovian limit and can be efficiently calculated based on the multi-time quantum measurements of the system. A heuristic algorithm for reconstructing the underlying open quantum dynamics is proposed, whose complexity is directly related to the proposed non-Markovianity measures. The non-Markovianity measures and the reconstruction algorithm are demonstrated with numerical examples, together with a careful reexamination of the non-Markovianity in quantum dephasing dynamics.



# 1  Introduction

A non-Markovianity measure for a (classical or quantum) dynamical process quantifies the causal dependence of the current observation on the history events. For classical stochastic processes, non-Markovianity has been well understood with the help of the $\epsilon$-machine, which is an important instance of hidden Markov models that constructs a minimal set of *causal states* together with a causal transition tensor to reproduce the observed dynamical process [1, 2]. In the quantum case, a large number of (non-)Markovianity criteria have been proposed from various aspects [3–13] (see [14, 15] for reviews), but none with a clear operational interpretation or a clear correspondence with the classical limit. The issue is resolved in a seminal work [16] which proposes a sufficient and necessary criterion for a quantum process to be Markovian, together with a quantitive measure of the non-Markovianity, under the recently proposed *process tensor framework* [17, 18].

This operational Markov criterion can be understood as follows. We consider a multi-time quantum measurement on a quantum system: one first performs a time-ordered sequence of quantum operations on the system, where each quantum operation contains a measurement (which disentangles the system and the rest of the world) followed by a preparation of a new quantum state, and then one performs a tomography of the final state of the system. The quantum dynamics of the system is characterized to be Markovian if and only if the final state only depends on the last preparation. The central idea of this criterion is that to judge whether the underlying quantum dynamics is Markovian or not, one should look at its response against (multiple) interventions (quantum operations). Such a Markov criterion is also sufficiently general in that it makes no assumptions on the details of the quantum dynamics of the system.

Since the majority, if not all, of the quantum dynamics is non-Markovian, an at least equally important task is to have a quantitive non-Markovianity measure. Ref. [16] provides a conceptually the most general non-Markovianity measure for open quantum dynamics based on their proposed Markov criterion: the distance between the quantum process and the Markovian quantum process closest to it. However, explicit calculations based on it could be extremely difficult even for very simple cases. In a previous work [19], we define the *memory complexity* which characterizes the minimal size of the unknown environment such that the system plus environment undergoes unitary dynamics as a whole and that the reduced dynamics of the system is identical to the observed dynamics (the open quantum evolution (OQE) model [20]), which is directly inspired from the $\epsilon$-machine. The memory complexity can be efficiently calculated for bounded environment, however, it is not fully reconciled with the operational Markov criterion: it vanishes if and only if the system itself undergoes unitary dynamics, but could increase unboundedly over time even if the system undergoes Markovian quantum dynamics described by a general quantum map. This is because in the latter case one may still need an exponentially large environment to ensure that the system plus environment undergoes global unitary dynamics. Ideally, the (non-)Markovianity of a quantum process should only rely on the observed dynamics but not on the mechanisms behind it. There could exist circumstances (for example with classical noises) for which there are more efficient "quantum hidden Markov models" compared to the OQE model to generate the same observed quantum dynamics. In such cases, the memory complexity may not be a good non-Markovianity mea-

sure since it specifically means the complexity of reproducing the observed quantum dynamics purely quantum mechanically.

In this work, we propose two non-Markovianity measures for general open quantum dynamics which are fully compatible with the operational Markov criterion, together with a heuristic algorithm to reconstruct the (unknown) open quantum dynamics whose complexity is closely related to the proposed non-Markovianity measures. They can be seen as complementary of the memory complexity to better incorporate those situations where the OQE model may not be the most efficient. Technically, the proposed non-Markovianity measures are based on a more general assumption compared to the OQE model, that is, the system plus environment undergoes Markovian quantum dynamics (MQE) described by an (unknown) quantum map $\mathcal{E}$. Here we note that the OQE model is the most general description of arbitrary quantum dynamics (but it may not be the most efficient). Therefore the MQE modeling of the system-environment (SE) dynamics is not a necessity from fundamental physics, but is for practical convenience (efficiency). Particularly, for dissipative quantum dynamics, the OQE model would generally require an exponentially large environment, while the equivalent MQE model may only involve a few degrees of freedom (DOFs). Therefore the complexity of the experimental reconstruction of the open quantum dynamics (reconstructing the quantum hidden Markov model) could be drastically reduced if the MQE model is used instead of the OQE model. One could also think of an engineered situation where one can directly control and measure the environment (for example an engineered environment made of controllable qubits), in which case the SE dynamics is naturally described by a quantum map.

In the following we will first introduce the generalized process tensor framework based on the MQE model of the SE dynamics in Sec. 2. In particular we will show that the three important physical requirements for the process tensor: linearity, complete positivity (CP) and containment [18], are still satisfied in the generalized case. We then present the two non-Markovianity measures and discuss their physical significances in Sec. 3, complemented with a heuristic machine learning algorithm to reconstruct the unknown open quantum dynamics (the hidden MQE model). The reconstruction algorithm and the proposed non-Markovianity measures are demonstrated with numerical examples, together with a careful reexamination of the non-Markovianity in quantum dephasing dynamics in Sec.4. We conclusion in Sec.5.

## 2 Process tensor framework for MQE

The original definition of the process tensor framework is based on the OQE description of the SE dynamics [18]. Here we briefly review the main ideas of the process tensor framework and show that the process tensor framework can be straightforwardly generalized to the case that the SE dynamics is described by the MQE model, with all of its important physical properties still satisfied.

Traditionally, the study of open quantum dynamics often follows a top down approach, that is, one starts from a microscopic model which describes the unitary evolution of the system coupled to an environment, and then obtains the reduced dynamics of the system by tracing out the environment [21], or that one directly starts from some phenomenological quantum master equations which focus on the system dynamics only [20, 22]. In either description, the open quantum dynamics of the system is known in priori, at least in principle. With the rapid progresses of quantum computing and quantum simulation technologies [23–27], the top down approach alone is no longer satisfying, since the noises on those quantum devices could be extremely difficult to know before hand. In the mean time, there is an increasing need for a quantitive description of the noises on near-term quantum devices, such as to characterize the overall fidelities of noisy quantum experiments, or for error correction and error mitiga-

tion [28–31]. Therefore an efficient way to characterize the (non-Markovian) open quantum dynamics based only on the experimentally accessible quantities, instead of resorting to the top down approach, is highly desirable.

Experimentally, one is often able to intervene the system dynamics by performing a quantum operation to prepare some initial state for the system at a time $t = 0$ (denoted as $\rho_0^S$), and then record the response by performing a tomography of the system state at a later time $t = \Delta$, a standard procedure known as quantum process tomography [32, 33]. If the quantum dynamics of the system is unitary or more generally Markovian, then it could be fully characterized by the reconstructed quantum map between the system states at time 0 and $\Delta$, denoted as $\mathcal{E}_{\Delta:0}^S$ [34, 35]. However, for non-Markovian quantum dynamics, this is not enough since in general $\mathcal{E}_{k\Delta:0}^S \neq (\mathcal{E}_{\Delta:0}^S)^k$ (Even worse, in the non-Markovian case even if this equation holds, $\mathcal{E}_{\Delta:0}^S$ alone is still insufficient to fully characterize the underlying quantum dynamics as will be shown later). In such case additional information is required to fully characterize the open quantum dynamics of the system. Fortunately, the quantum map does not describe all the probes one could possibly perform on the system either. For example, one could intervene the system dynamics by performing two quantum operations at two times $t_0$ and $t_1$, and then measure the output state at time $t_2$ (a three-time quantum measurement). In fact, all such three-time quantum measurements constitute a two-step *process tensor*. Generally, a $k$-step process tensor is defined as a multilinear map from $k$ quantum operations, denoted as $\Lambda_{k-1:0} = \{\Lambda_0, \Lambda_1, \ldots, \Lambda_{k-1}\}$ at $k$ different times $\{t_0, t_1, \ldots, t_{k-1}\}$, to the output quantum state $\rho_k$ at time $t_k$, which is

$$\rho_k = \mathrm{tr}_E\left(\mathcal{E}_{k:k-1}\Lambda_{k-1}\mathcal{E}_{k-1:k-2}\ldots\Lambda_1\mathcal{E}_{1:0}\Lambda_0\rho_0^{SE}\right). \tag{1}$$

Here $\rho_0^{SE}$ is the SE initial state. Each $\Lambda_j$ is itself a CP quantum map of size $d^2 \times d^2$ ($d$ is the Hilbert space size of the system), and can generally be implemented as a quantum measurement $M_j$ followed by a preparation $P_j$, each of size $d \times d$ [36]. $\mathcal{E}_{j:j-1}$ denotes the SE evolutionary operator from time step $j-1$ to $j$ which is a complete positive and trace preserving (CPTP) quantum map. The operation of $\mathcal{E}$ (subscripts for time steps are omitted for briefness if they are not necessary) on an input SE state $\rho^{SE}$ can generally be written in the Sudarshan-Kraus-Choi form: $\mathcal{E}(\rho^{SE}) = \sum_s A_s \rho^{SE} A_s^\dagger$, with the normalization condition $\sum_s A_s^\dagger A_s = I$ ($I$ is the identity matrix) [34, 37, 38]. Explicitly, we denote

$$\mathcal{E}(\rho^{SE}) = \sum_{s,i,\alpha,i',\alpha'} A_{s,o,\beta,i,\alpha}\rho_{i,\alpha,i',\alpha'}^{SE}A_{s,o',\beta',i',\alpha'}^*$$
$$= \sum_{i,i',\alpha,\alpha'} \rho_{i,\alpha,i',\alpha'}^{SE}W_{\alpha,\alpha',\beta,\beta'}^{i,i',o,o'}, \tag{2}$$

where $i, i', o', o$ are the system indices and $\alpha, \alpha', \beta, \beta'$ are the environment indices. In Eq.(2) we have also used

$$W_{\alpha,\alpha',\beta,\beta'}^{i,i',o,o'} = \sum_s A_{s,o,\beta,i,\alpha}A_{s,o',\beta',i',\alpha'}^*, \tag{3}$$

as the matrix representation of $\mathcal{E}$. The normalization condition can be simply read as

$$\sum_{o,\beta} W_{\alpha,\alpha',\beta,\beta}^{i,i',o,o} = \delta_{i,i'}\delta_{\alpha,\alpha'}. \tag{4}$$

Eq.(1) implicitly defines the $k$-step process tensor $\Upsilon_{k:0}$: $\rho_k = \Upsilon_{k:0}(\Lambda_{k-1:0})$ (explicit matrix representations are used for $\mathcal{E}_{j:j-1}$ and $\Lambda_j$ throughout the text), which is shown in Fig. 1(a,b). In case $\mathcal{E}_{j:j-1}$ is unitary, Eq.(1) naturally reduces to the original definition in Ref. [18]. The process tensor $\Upsilon_{k:0}$ is a natural extension of the quantum map defined at two times (preparation

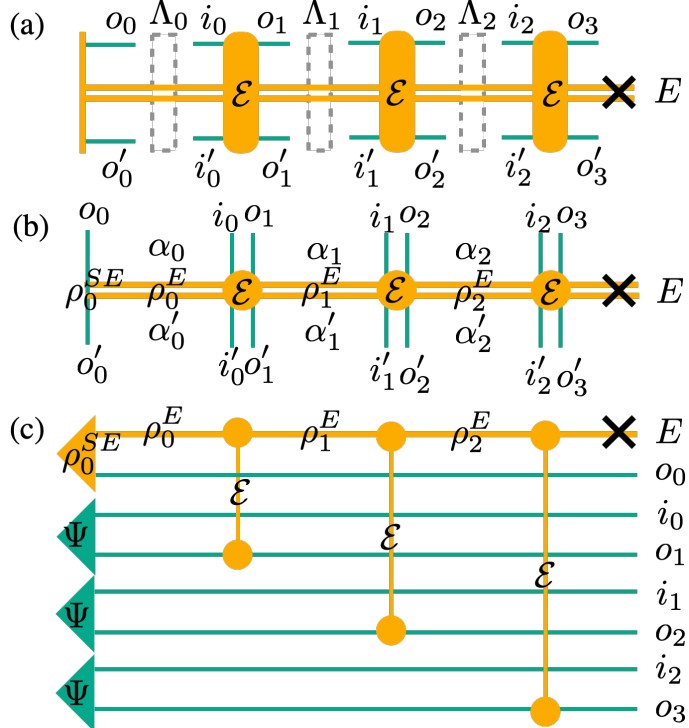

Figure 1: (a) Demonstration of a 3-step process tensor. $\mathcal{E}$ is the system-environment evolutionary operator and $\rho_0^{SE}$ is the SE initial state. $i_j, o_j$ denote the input and output indices at the $j$-th time step ($o_0$ corresponds to the initial state of the system). $\Lambda_j$ denotes the $j$-th quantum operation which could be inserted after the $j$-th time step to intervene the system dynamics. (b) The Matrix Product Operator representation of the process tensor. (c) The quantum circuit implementation of the process tensor as a many-body quantum state, where $\Psi$ is the maximally entangled state. In panels (b,c) $\rho_j^E$ means the effective environment state after time step $j$ defined in Eq.(16). The environment is traced out in all the panels in the end.

at $t_0$ and measurement at $t_1$) to multiple times. In fact it describes the most general observations one could possibly make on the system. Compared to the classical case, we can see that it plays the role of the conditional probability $P(x_k|x_{k-1},\ldots,x_0)$, which fully characterizes a classical stochastic process.

The process tensor is naturally a matrix product operator (MPO) as shown in Fig. 1(b), which can be written as:

$$\Upsilon^{o_0',i_0',o_1',\ldots,i_{k-1}',o_k'}_{o_0,i_0,o_1,\ldots,i_{k-1},o_k} = \sum_{\alpha_{k-1:0},\alpha_{k-1:0}',\alpha_k} W^{o_0,o_0'}_{\alpha_0,\alpha_0'} W^{i_0,i_0',o_1,o_1'}_{\alpha_0,\alpha_0',\alpha_1,\alpha_1'} \times \cdots \times W^{i_{k-1},i_{k-1}',o_k,o_k'}_{\alpha_{k-1},\alpha_{k-1}',\alpha_k,\alpha_k}, \tag{5}$$

with $i_j, i_j', o_j, o_j'$ the "physical indices" and $\alpha_j, \alpha_j'$ the "auxiliary indices" (environmental indices). We have also used $\alpha_{k-1:0}$ to denote the set of indices $\{\alpha_0, \alpha_1, \ldots, \alpha_{k-1}\}$ (and similarly for $\alpha_{k-1:0}'$). $W^{o_0,o_0'}_{\alpha_0,\alpha_0'}$ is the SE initial state $(\rho_0^{SE})^{o_0,o_0'}_{\alpha_0,\alpha_0'}$. Moreover, $\Upsilon_{k:0}$ is a Matrix Product Density Operator (MPDO) [39,40], which is a special form of MPO that guarantees positivity by construction, since each site tensor $W$ is positive by Eq.(3).

Interestingly, it is pointed out that $\Upsilon_{k:0}$ can be implemented using a quantum circuit as shown in Fig. 1(c) [18], where $\Psi = \frac{1}{d}\sum_{m,n=1}^d |m\rangle_o \langle n|_o \otimes |m\rangle_i \langle n|_i$ is the maximally entangled state (the subscripts indicate that they correspond to the input or output indices). This can be

seen by looking at the effects of each pair of the SE interaction in Fig. 1(c):

$$\mathcal{E}(\Psi \otimes |\alpha\rangle\langle\alpha'|) = \frac{1}{d}\sum_{m,n=1}^{d}|m\rangle_i\langle n|_i \otimes \mathcal{E}(|m\rangle_o\langle n|_o \otimes |\alpha\rangle\langle\alpha'|)$$

$$= \frac{1}{d}\sum_{m,n,m',n',\beta,\beta'} W_{\alpha,\alpha',\beta,\beta'}^{m,n,m',n'}|m\rangle_i\langle n|_i \otimes |m'\rangle_o\langle n'|_o \otimes |\beta\rangle\langle\beta'|, \tag{6}$$

where the last line is exactly the matrix representation of $\mathcal{E}$ in Eq.(3), up to a normalization constant $1/d$. As a result the output quantum state of the quantum circuit in Fig. 1(c) generates $\Upsilon_{k:0}$ up to an overall normalization constant $(1/d)^k$ after tracing out the environment index in the end. Using this quantum circuit, the process tensor defined at multiple times is mapped into a multi-qubit quantum state. Therefore instead of doing a multi-time quantum process tomography, one could perform a multi-qubit quantum state tomography to obtain the process tensor.

In case $\mathcal{E}$ is a unitary operation, the output of the quantum circuit in Fig. 1(c) is a sequentially generated multi-qubit state for which a polynomial tomography algorithm against $k$ exists for bounded environment [19]. Unfortunately, for general $\mathcal{E}$, no efficient tomography algorithm with guaranteed convergence exists to our knowledge, even if the environment is bounded. This is because that the existing deterministic and efficient tomography algorithms based on Matrix Product States (MPSs) only work if the underlying mixed quantum states are pure or *fairly pure* [41–43], namely they can be written as the sum of a few pure states [44] (thus with entropy $S \propto O(1)$), while an MPDO can easily represent a mixed quantum state whose entropy grows as a volume law ($S \propto O(k)$). Nevertheless, for bounded environment the unknown $\Upsilon_{k:0}$ can be efficiently parameterized using only a polynomial number of parameters as in Eq.(5), and in practice one can use a variational MPO [45,46] or MPDO [47] ansatz in combination with a machine learning algorithm to efficiently reconstruct the process tensor in those forms.

Now we show that the three important properties of the process tensor: linearity, complete positivity and containment, are still satisfied for general $\mathcal{E}$. The linearity condition is trivially satisfied, since Eq.(1) is linear against each of the input $\Lambda_j$. The CP condition is equivalent to require that $\Upsilon_{k:0}$ in Eq.(5) is positive. To see this, one can define the tensor

$$\mathcal{A}_{s_{k:0},o_{k:0},i_{k-1:0},\alpha_k} = \sum_{\alpha_{k-1:0}} A_{s_0,o_0,\alpha_0}A_{s_1,o_1,\alpha_1,i_0,\alpha_0} \times \dots A_{s_k,o_k,\alpha_k,i_{k-1},\alpha_{k-1}}, \tag{7}$$

with $\rho_0^{SE} = \sum_{s_0} A_{s_0,o_0,\alpha_0}A^*_{s_0,o_0,\alpha_0}$, then the operation of $\Upsilon_{k:0}$ on $\Lambda_{k-1:0}$ can be rewritten as

$$(\rho_k)_{o_k,o'_k} = \Upsilon_{k:0}(\Lambda_{k-1:0}) =$$
$$\sum_{s_{k:0},\alpha_k,o_{k:0},i_{k-1:0},o'_{k:0},i'_{k-1:0}} \mathcal{A}_{s_{k:0},o_{k:0},i_{k-1:0},\alpha_k}\Lambda_{o_0,o'_0,i_0,i'_0} \times \dots \times \Lambda_{o_{k-1},o'_{k-1},i_{k-1},i'_{k-1}}\mathcal{A}^*_{s_{k:0},o'_{k:0},i'_{k-1:0},\alpha_k}, \tag{8}$$

which is indeed in the standard Sudarshan-Kraus-Choi form. The containment condition ensures that any measurement outcome within $k$ time steps is independent of the quantum operations made after $k$, defined as

$$\text{tr}_{o_k}(\Upsilon_{k:0}) = \delta_{i_{k-1},i'_{k-1}} \otimes \Upsilon_{k-1:0}, \tag{9}$$

which can be easily proven by substituting Eq.(5) into Eq.(9) and using the normalization condition in Eq.(4).

# 3 Non-Markovianity measures

In Ref. [16], the non-Markovianity of a $k$-step quantum process is defined as the distance (any CP-contractive quasi-distance) between the actual process tensor $\Upsilon_{k:0}$ and the Markovian process tensor, denoted as $\Upsilon_{k:0}^{\mathrm{Markov}}$, closest to it. This non-Markovianity measure, although conceptually the most general one, is impractical for actual evaluation, because: 1) identification of the closest Markovian process tensor to a given process tensor may not be straightforward and 2) even if the closest Markovian process tensor is identified, computing the distance between it and the given process tensor may still be very hard and not scalable.

Ideally, a non-Markovianity measure should meet the following requirements: 1) It should be uniquely defined, that is, it should only depend on the physically measurable quantities (the process tensor), instead of the hidden OQE or MQE model; 2) It should be intuitive and trivially predicts the Markovian limit; 3) It should be *easily calculable*, and describes the complexity of reconstructing the underlying open quantum dynamics.

Here we propose two non-Markovianity measures, based on the MPO and MPDO representations of the process tensor respectively. The reason that we consider these two representations is because they both are commonly used for representing positive operators and there is no decisive advantage of one over another. For example, MPO could be more efficient in terms of the number of parameters required to represent a quantum operator. In fact if a (positive) quantum operator can be efficiently represented as an MPDO, then it can also be efficiently represented as an MPO, while the reverse is not true in general [48]. Additionally, an MPDO can easily be converted into an MPO and the reverse is also not true. However, an MPO representation of the process tensor does not guarantee positivity, therefore inaccuracy during the process tensor tomography would likely result in an unphysical process tensor if an MPO ansatz is used. This also represents a complication compared to the MPS representation of pure states, in the latter case there is a standard canonical form of MPS (the mixed-canonical form) to be used for variational optimization [49]. The first non-Markovianity measure we propose, which corresponds to the MPO representation, is based on the operator space entanglement entropy (OSEE) [50]. And the second non-Markovianity measure is based on the entanglement entropy of an *effective environment state* inspired from the memory complexity in the unitary case [19]. In the next we will elaborate on both non-Markovianity measures.

## 3.1 OSEE as non-Markovianity measure

A quantum operator in MPO form can be treated similarly as an MPS by vectorizing it into a pure state, which means the mapping

$$|\Upsilon\rangle_{o_0,p_0,i_0,q_0,o_1,p_1,\dots,i_{k-1},q_{k-1},p_k} \longleftrightarrow \Upsilon^{p_0,q_0,p_1,\dots,q_{k-1},p_k}_{o_0,i_0,o_1,\dots,i_{k-1},o_k}. \tag{10}$$

The OSEE at time step $j$, denoted as $S_j^o$, is defined as the bipartition entanglement entropy of $|\Upsilon\rangle$ by splitting it into two subsystems, one contains all the indices before (and include) $j$ and the other contains the rest. The non-Markovianity measure based on the OSEE of the process tensor is defined as

$$\mathcal{N}_j^{osee} = \frac{1}{2}S_j^o = \frac{1}{2}S\left(\mathrm{tr}_{i_{k-1:j},q_{k-1:j},o_{k:j+1},p_{k:j+1}}\left(|\Upsilon\rangle\langle\Upsilon|\right)\right), \tag{11}$$

where $S(\rho) = \rho\log_2(\rho)$ is the entanglement entropy of $\rho$ (The more general quantum Renyi entropy $\log_2(\mathrm{tr}(\rho^\alpha))/(1-\alpha)$ can also be used). The factor $1/2$ is added such that $\mathcal{N}_j^{osee}$ reduces to the memory complexity when $\mathcal{E}$ is unitary. $\mathcal{N}_j^{osee}$ is uniquely defined since the bipartition entanglement entropy of $|\Upsilon\rangle$ is uniquely defined, and it can certainly be efficiently calculated given the MPO form of $\Upsilon$ as a standard practice for MPO [49]. Moreover, it straightforwardly

predicts the Markovian limit, namely the underlying quantum process is Markovian if and only if $\mathcal{N}_j^{osee} = S_j^o = 0$ for all $j$ considered. The is because $S_j^o = 0$ means that $\Upsilon$ is separable, which is exactly the necessary and sufficient condition for the Markovian limit [16].

There is a *boundary effect* when using $\mathcal{N}^{osee}$ as the non-Markovianity measure, which is shown as follows. For a $k$-step quantum process with a separable SE initial state, $\mathcal{N}_j^{osee}$ is defined for $1 \leq j < k$ which satisfies $\mathcal{N}_j^{osee} \leq \log_2(d^{2j})$ and $\mathcal{N}_{k-j}^{osee} \leq \log_2(d^{2j})$ (Starting from the boundaries, the bond dimension, namely the size of the auxiliary index of $|\Upsilon\rangle$ can not grow faster than $d^{2j}$ since it is an MPS with open boundary condition and with physical dimension $d^2$). The first inequality is due to the ignorance of the initial state (so that the best we can do is to start from some unitary SE initial state as will be shown in the reconstruction algorithm for the open quantum dynamics). While the second inequality is purely an unphysical boundary effect due to a finite value of $k$. For example, $\mathcal{N}_j^{osee}$ may decrease when $j$ approaches $k$, and if one considers a $k+1$-step process instead one could get a very different value for those $\mathcal{N}_j^{osee}$ with $j$ close to $k$. The boundary effect can be eliminated by considering a large $k$ and focusing on those $\mathcal{N}_j^{osee}$ with $j$ far away from the right boundary $k$.

## 3.2 Entanglement entropy of an effective environment state as non-Markovianity measure

The memory complexity of a quantum process at a time step $j$, denoted as $\mathcal{C}_j$, is defined as the entanglement entropy of an effective environment state $\rho_j^E$ (the state shown in Fig. 1(c) in case $\mathcal{E}$ is unitary), which contains all the history information (with time steps not greater than $j$) such that the outcome of any future quantum operations after $j$ only depends on $\rho_j^E$ [19]. $\rho_j^E$ thus acts like a memory state, which is closely related to the causal states of the $\epsilon$-machine, as well as the memory state of the q-simulator and the infinite MPS descriptions for classical stochastic processes [51–54]. $\mathcal{C}_j$ is also the bipartition entanglement entropy between the $j$-step process tensor $\Upsilon_{j:0}$ and the environment state $\rho_j^E$ (These two subsystems form a pure state as a whole in the OQE model, thus the bipartition entanglement entropy is well defined [19]). Formally, the second non-Markovianity measure is defined in the same way as the memory complexity, namely

$$\mathcal{N}_j^{ee} = S(\rho_j^E), \tag{12}$$

where the effective environment state $\rho_j^E$ for a general non-unitary $\mathcal{E}$ is constructed in the following.

We first consider the expectation value of a sequence of quantum operations with $j$ quantum operations followed by a quantum measurement in the end, denoted as $\{\Lambda_0, \dots, \Lambda_{j-1}, M_j\} = \{M_0, P_0, \dots, M_{j-1}, P_{j-1}, M_j\}$, which can be computed as

$$\operatorname{tr}(M_0 \otimes P_0 \otimes M_1 \otimes \cdots \otimes P_{j-1} \otimes M_j \Upsilon_{j:0}) = \sum_{i_{j-1:0}, i'_{j-1:0}, o_{j:0}, o'_{j:0}, \alpha_{j-1:0}, \alpha'_{j-1:0}, \alpha_j} \left( W^{o_0, o'_0}_{\alpha_0, \alpha'_0} M_0^{o_0, o'_0} \right)$$

$$\times \left( P_0^{i_0, i'_0} W^{i_0, i'_0, o_1, o'_1}_{\alpha_0, \alpha'_0, \alpha_1, \alpha'_1} M_1^{o_1, o'_1} \right) \times \cdots \times \left( P_{j-1}^{i_{j-1}, i'_{j-1}} W^{i_{j-1}, i'_{j-1}, o_j, o'_j}_{\alpha_{j-1}, \alpha'_{j-1}, \alpha_j, \alpha_j} M_j^{o_j, o'_j} \right), \tag{13}$$

where the site tensors of $\Upsilon$ with time steps larger than $j$ are not required due to the containment condition. Then we define the expectation value of a "local" quantum measurement $M_j$ as the average over all the past quantum operations $\{M_0, P_0, \dots, M_{j-1}, P_{j-1}\}$ except for $M_j$,

which is denoted as $\mathrm{tr}(M_j \Upsilon_{j:0})$ and can be computed by

$$
\mathrm{tr}(M_j \Upsilon_{j:0}) = \sum_{i_{j-1:0}, o_{j-1:0}, o_j, o'_j, \alpha_{j-1:0}, \alpha'_{j-1:0}, \alpha_j} W^{o_0,o_0}_{\alpha_0,\alpha'_0} W^{i_0,i_0,o_1,o_1}_{\alpha_0,\alpha'_0,\alpha_1,\alpha'_1} \times \dots
$$
$$
\times W^{i_{j-2},i_{j-2},o_{j-1},o_{j-1}}_{\alpha_{j-2},\alpha'_{j-2},\alpha_{j-1},\alpha'_{j-1}} \left( W^{i_{j-1},i_{j-1},o_j,o'_j}_{\alpha_{j-1},\alpha'_{j-1},\alpha_j,\alpha_j} M^{o_j,o'_j}_j \right). \tag{14}
$$

To this end, we note that the "local" expectation $\mathrm{tr}(M_j \Upsilon_{j:0})$ we have defined above is very distinct from the case that one *does nothing* in time steps from 0 to $j-1$ and only performs a measurement $M_j$ at the $j$-th time step. Mathematically, the latter means a very different way for tensor contraction:

$$
\mathrm{tr}\left(M_j \mathcal{E}_{j:j-1} \mathcal{E}_{j-1:j-2} \dots \mathcal{E}_{1:0} \rho_0^{SE}\right) = \sum_{o_{j:0}, o'_{j:0}, \alpha_{j-1:0}, \alpha'_{j-1:0}, \alpha_j} W^{o_0,o'_0}_{\alpha_0,\alpha'_0} W^{o_0,o'_0,o'_1,o'_1}_{\alpha_0,\alpha'_0,\alpha_1,\alpha'_1} \times \dots
$$
$$
\times W^{o_{j-2},o'_{j-2},o_{j-1},o'_{j-1}}_{\alpha_{j-2},\alpha'_{j-2},\alpha_{j-1},\alpha'_{j-1}} \left( W^{o_{j-1},o'_{j-1},o_j,o'_j}_{\alpha_{j-1},\alpha'_{j-1},\alpha_j,\alpha_j} M^{o_j,o'_j}_j \right). \tag{15}
$$

Physically, the former means that we do make preparations and measurements at all the past time steps and then average over them, while the latter means that we only make a quantum measurement at time step $j$.

From Eq.(14) we can see that if we define the effective environment state $\rho_{j-1}^E$ after time step $j-1$ as

$$
(\rho_{j-1}^E)_{\alpha_{j-1},\alpha'_{j-1}} = \sum_{i_{j-2:0}, o_{j-1:0}, \alpha_{j-2:0}, \alpha'_{j-2:0}} W^{o_0,o_0}_{\alpha_0,\alpha'_0} \times W^{i_0,i_0,o_1,o_1}_{\alpha_0,\alpha'_0,\alpha_1,\alpha'_1} \dots W^{i_{j-2},i_{j-2},o_{j-1},o_{j-1}}_{\alpha_{j-2},\alpha'_{j-2},\alpha_{j-1},\alpha'_{j-1}}, \tag{16}
$$

then $\mathrm{tr}(M_j \Upsilon_{j:0})$ can be simply computed as

$$
\mathrm{tr}(M_j \Upsilon_{j:0}) = \mathrm{tr}\left(M_j \mathcal{E}_{j:j-1}(I^S \otimes \rho_{j-1}^E)\right), \tag{17}
$$

with $I^S$ the identity matrix of the system. Therefore to compute the local expectation value of $M_j$, or more generally any observables beyond (include) time step $j$, all one needs is the $\rho_{j-1}^E$ from the past. $\rho_j^E$ can also be recursively computed as

$$
\rho_j^E = \mathrm{tr}_S\left(\mathcal{E}_{j:j-1}(I^S \otimes \rho_{j-1}^E)\right), \tag{18}
$$

with $\rho_0^E = \mathrm{tr}_S(\rho_0^{SE})$. The $\rho_j^E$ defined in this way plays a similar role to the distribution of the causal states in the $\epsilon$-machine [55], thus we define Eq.(12) as the second non-Markovianity measure. In case $\mathcal{E}$ is unitary, $\rho_j^E$ reduces to the original definition in Ref. [19].

Compared to $\mathcal{N}^{osee}$, $\mathcal{N}^{ee}$ is more physically motivated, and is free of the boundary effect. It is also easy to be calculated given the "correct" MPDO form of $\Upsilon$ and it is straightforward to see that the quantum process is Markovian if and only if $\mathcal{N}_j^{ee} = 0$. However, the MPDO form of $\Upsilon$ (reconstructed from experiment) is not uniquely defined, and if one obtains an MPO form of $\Upsilon$ or even the exact $\Upsilon$ from the experiment, there is no unique way to decompose it into an MPDO [56]. Moreover, a general MPDO does not have to satisfy the normalization condition for each of its site tensor, which means that to evaluate Eq.(14) one needs both the site tensors from the past and future! Eqs.(4,5) actually define some "canonical form" of MPDO (a generic MPDO does not require Eq.(4) to be satisfied). This canonical form could be ensured in practice by a carefully designed MPDO ansatz which also satisfies Eq.(4) by construction. The canonical form of MPDO is not unique either. To see this, we look at the transformation of $\rho_j^E$ under a basis change for the environment, denoted as $\Lambda$ ($\Lambda$ does not

have to be unitary), which maps the original environment with size $D$ to a new environment $E'$ with size $D'$ ($D' \geq D$). Under this basis change, we have

$$\rho^{SE'} = \Lambda \rho^{SE}, \tag{19}$$

$$\mathcal{E}' = \Lambda \mathcal{E} \Lambda^{-1}, \tag{20}$$

where $\Lambda^{-1}$ is understood as the Moore–Penrose pseudo-inverse in case $D < D'$. Substituting Eqs.(19, 20) into Eq.(16), we get $\rho_j^{E'}$ in the new basis

$$\rho_j^{E'} = \Lambda \rho_j^E. \tag{21}$$

The basis transformation has no observable effects since the environment will be traced out in the end (again using Eq.(4)), however it *does affect* $\mathcal{N}^{ee}$ since $\rho_j^{E'}$ would generally be different from $\rho_j^E$.

To resolve this issue with $\mathcal{N}^{ee}$, we consider the realistic situation where one wants to reconstruct the open quantum dynamics by assuming a hidden MQE model and reconstructing this model. *A priori* one has no knowledge of the SE initial state and the best one can do is to assume a simplest choice of it: a pure state with zero entropy (more details will be shown later in the tomography algorithm). Such a choice would select a particular environment basis with a small $\mathcal{N}^{ee}$ in practice. Whether such a choice gives theoretically the smallest $\mathcal{N}^{ee}$ out of all the possible cases related by Eq.(21) is left to further investigation.

### 3.3 A heuristic algorithm for reconstructing the open quantum dynamics

The process tensor describes all the measurable quantities and fully characterizes the open quantum dynamics. However it would be impractical to reconstruct the process tensor for arbitrarily large $k$ experimentally. Therefore similar to the classical case [2], one would like to reconstruct a *hidden Markov model* (similar to the $\epsilon$-machine) based on a limited number of observations to characterize the underlying quantum process. In this sense, the ultimate goal of process tensor tomography would be to reconstruct the hidden OQE or MQE model, instead of obtaining the process tensor itself. This is also the reason why different assumptions of the model for the SE dynamics may significantly affect the complexity of reconstructing the open quantum dynamics.

The non-Markovianity measures $\mathcal{N}^{osee}$ and $\mathcal{N}^{ee}$ are directly related to the complexity of reconstructing the hidden MQE model, in terms of the least number of unknown parameters needs to be fixed by the reconstruction algorithm. Here we present a variational algorithm for reconstructing the hidden MQE model, under the assumptions that 1) $\mathcal{E}$ is time-independent; 2) the environment size is bounded by some integer $D$; 3) the size of the internal auxiliary index $s_j$, $\dim(s_j)$, is bounded by an integer $R$ ($R$ is naturally bounded by $d^2 D^2$).

Under these assumptions we can parameterize each site tensor $W$ using a single parametric tensor $\bar{A}_{s,o,\beta,i,\alpha}$ which contains $d^2 D^2 R$ variational parameters. To predict a $k$-step process tensor denoted by $\bar{\Upsilon}_{k:0}$, one still needs to fix the initial state, for which we simply assume that the SE initial state is a pure state: $\bar{\rho}_0^{SE} = |\psi^{SE}\rangle\langle\psi^{SE}|$. We note that this assumption does not loss any generality since if the initial state is mixed, one could purify it using external DOFs and enlarge $\mathcal{E}$ accordingly by tensor product with the identity matrix on those external DOFs [19]. The specific choice of SE initial state does not affect $\mathcal{N}^{osee}$ since $\mathcal{N}^{osee}$ can be computed purely based on $\Upsilon$. However, as shown in Eq.(21), it does affect $\mathcal{N}^{ee}$ but likely in a good direction that one could make use of this property to select a simple environment state with smaller (if not the smallest) $\mathcal{N}^{ee}$, while generating the equivalent open quantum dynamics. Now any $\bar{\Upsilon}_{k:0}$ can be predicted by substituting $\bar{\rho}_0^{SE}$ and $\bar{A}_{s,o,\beta,i,\alpha}$ into Eq.(5). Given an experimentally

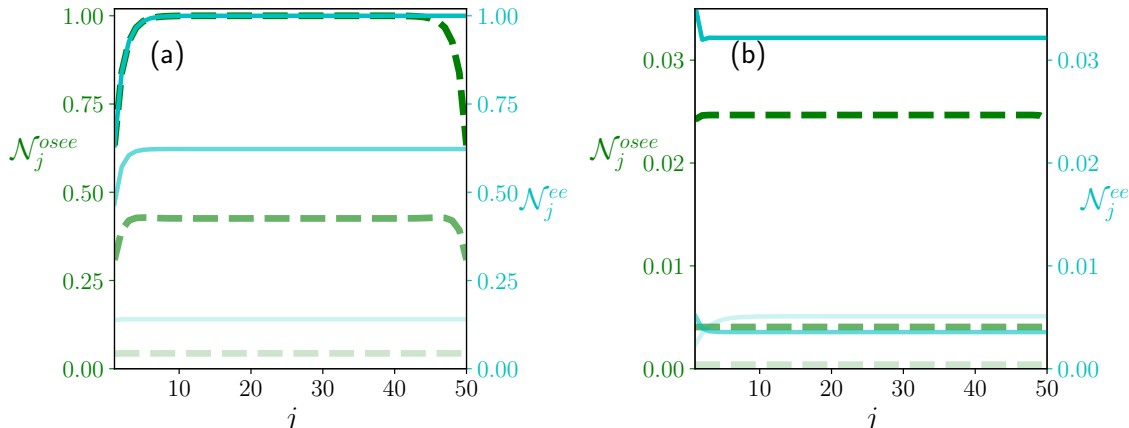

Figure 2: The two non-Markovianity measures, $\mathcal{N}_j^{osee}$ and $\mathcal{N}_j^{ee}$ as a function of the time step $j$ calculated for a dissipative two-spin XX chain with $k = 51$ (thus $1 \leq j \leq 50$). The green dashed lines (for $\mathcal{N}^{osee}$) and the cyan solid lines (for $\mathcal{N}^{ee}$) from top down (also from darker to lighter) correspond to $\Gamma = 0, 1, 5$ with $n = 0$ in (a) and to $\Gamma = 5, 10, 20$ with $n = 0.5$ in (b). Here we have used $\Delta = 0.3$.

reconstructed $k$-step process tensor $\Upsilon_{k:0}$, we can thus optimize $\bar{\rho}_0^{SE}$ and $\bar{A}_{s,o,\beta,i,\alpha}$ by minimizing the following loss function (one could of course use other loss functions)

$$\text{loss}(\bar{\rho}_0^{SE}, \bar{A}) = |\bar{\Upsilon}_{k:0} - \Upsilon_{k:0}|^2, \tag{22}$$

where $|\cdot|$ denotes the the square of the Euclidean norm. Once we have obtained the optimal $\bar{\rho}_0^{SE}$ and $\bar{A}$, both $\mathcal{N}_j^{osee}$ and $\mathcal{N}_j^{ee}$ can be efficiently computed for any $j$. Here we have not explicitly enforced Eq.(4) for $\bar{A}$, but we should be able to get a descent $\bar{A}$ that satisfies this condition if we do not loss too much precision during the optimization. In practice, one may also gradually enlarge $k$ in Eq.(22) until that the loss value does not fluctuate significantly when increasing $k$, so as to obtain a descent $\bar{\rho}_0^{SE}$ and $\bar{A}$ with a minimal experimental effort. One could also use a gradient-based optimization algorithm to accelerate the convergence by evaluating the gradient of Eq.(22) with automatic differentiation [57].

# 4  Examples

In the following we will show two examples with numerical simulations. In the first example we consider a dissipative two-spin XX chain, with which we demonstrate the behaviors of the two proposed non-Markovianity measures. In the second example we reexamine the quantum dephasing dynamics of a single spin to discuss the subtlety against whether it is Markovian or not.

Here we stress that although for bounded environment the complexities of evaluating the two non-Markovianity measures $\mathcal{N}^{osee}$ and $\mathcal{N}^{ee}$ (based on the MPO and MPDO representations of the process tensor respectively), as well as the heuristic algorithm introduced in Sec. 3.3 to reconstruct the hidden MQE, are all polynomial in the number of time steps $k$, they will grow exponentially with the system size for a many-body system. As such we will limit ourself to a single qubit system in our numerical examples.

## 4.1 A dissipative two-spin XX chain

To demonstrate the behaviors of the two proposed non-Markovianity measures, we consider the example of a two-spin XX chain with dissipative driving on the second spin. We treat the first spin as the system and the second as the environment. The overall SE dynamics is described by the Lindblad master equation [58, 59]

$$\frac{d\rho^{SE}}{dt} = \mathcal{L}(\rho^{SE}) = -\mathrm{i}[H_{XX}, \rho^{SE}] + \mathcal{D}(\rho^{SE}), \tag{23}$$

with the Hamiltonian $H_{XX} = J(\sigma_x^S \sigma_x^E + \sigma_y^S \sigma_y^E)$ (we set $J = 1$ as the unit), and the dissipator

$$\begin{aligned} \mathcal{D}(\rho^{SE}) =& \Gamma(1-n)\left(2\sigma_-^E \rho^{SE}\sigma_+^E - \{\sigma_+^E \sigma_-^E, \rho^{SE}\}\right) \\ &+ \Gamma n\left(2\sigma_+^E \rho^{SE}\sigma_-^E - \{\sigma_-^E \sigma_+^E, \rho^{SE}\}\right), \end{aligned} \tag{24}$$

which drives the environment spin towards a local steady state $\rho_{st}^E = (1-n)|0\rangle\langle 0| + n|1\rangle\langle 1|$ with a rate $2\Gamma$. The SE initial state is assumed to be $\rho_0^{SE} = \rho_0^S \otimes \rho_{st}^E$. The discrete-time quantum map for the system plus environment is $\mathcal{E} = \exp(\mathcal{L}\Delta)$ with $\Delta$ the time step size.

To show the behaviors of the two proposed non-Markovianity measures, we reconstruct the hidden MQE using the algorithm in Sec. 3.3, with $\Upsilon_{k:0}$ computed analytically, and then based on the reconstructed $\bar{\Upsilon}$ we can compute $\mathcal{N}^{osee}$ and $\mathcal{N}^{ee}$. Other than demonstrating the reconstruction algorithm, the reason why we do not directly compute $\mathcal{N}^{osee}$ and $\mathcal{N}^{ee}$ based on the exact $\Upsilon_{k:0}$ is that $\mathcal{N}^{ee}$ is dependent on the choice of the environment initial state, and it is more reasonable to choose a simple (pure) state for it during reconstruction due to the ignorance of the environment and also to select the simplest possible environment as discussed in Sec. 3.3. For the reconstruction, we have used $D = 2$ and $R = d^2 D^2 = 16$. The BFGS algorithm is used as the optimization solver, and during the optimization we have gradually increased $k$ from 2 to a maximum value of 6.

We study the dependencies of $\mathcal{N}^{osee}$ and $\mathcal{N}^{ee}$ (computed with $\bar{\Upsilon}$) on $\Gamma$ for $n = 0, 0.5$ respectively, with the results shown in Fig. 2. The case $n = 0$ is shown in Fig. 2(a), where the dissipator drives the environment towards the pure state $|0\rangle\langle 0|$. Interestingly, as shown in Ref. [19], without dissipation the unitary dynamics will drive the environment towards the maximally mixed state $I^E/2$, therefore there will be a competition between the unitary and dissipative dynamics. We can clearly see from Fig. 2(a) that for $\Gamma = 0$, the unitary dynamics wins and $\mathcal{N}_j^{osee} \approx \mathcal{N}_j^{ee} \approx 1$ (noticing the boundary effect for $\mathcal{N}_j^{osee}$), and that for $\Gamma = 5$, the dissipative dynamics wins and we have both measures close to 0. In Fig. 2(b) we show the case $n = 0.5$, for which both the unitary and dissipative dynamics drive the environment towards $I^E/2$. In this case the loss function in Eq.(22) fails to converge to descent precision with small $\Gamma$s (ideally the loss should be 0 but the actual loss obtained after a large number (10000) of iterations still fails to converge to 0). Therefore we only show results for $\Gamma = 5, 10, 20$. For large $\Gamma$, the scales of the system and environment dynamics are well separated and one expects in this case that the adiabatic elimination is a good approximation, namely $\rho^{SE} \approx \rho^S \otimes \rho^E$, and an effective Lindblad equation for the system dynamics alone can be derived. Therefore the quantum dynamics of the system should be close to Markovian for large $\Gamma$, which is indeed the observed case (both non-Markovianity measures are small). Interestingly, even if the actual environment state is maximally mixed in the latter case, with a small value of $\mathcal{N}^{ee}$ it means that one could identify an effective environment state with much smaller entanglement entropy but generates the equivalent open quantum dynamics.

## 4.2 Reexamining the quantum dephasing dynamics

In case the open quantum dynamics of a quantum system can be described by the Lindblad master equation, it is usually characterized as Markovian or directly used as the definition of

Markovianity [5, 6]. However, it is argued that this may not be true [16, 60, 61], with the quantum dephasing dynamics of a single spin as an outstanding counter-example. Here we will carefully reexamine this discrepancy with intuition in the following.

The quantum dephasing dynamics of a single spin can be described by the following Lindblad master equation

$$\frac{d\rho^S}{dt} = \mathcal{L}^{dp}(\rho^S) = \gamma\left(\sigma_z^S \rho^S \sigma_z^S - \rho^S\right), \tag{25}$$

where $\gamma$ is the dephasing rate. Under Eq.(25) the diagonal terms of $\rho^S$ will not change while the off-diagonal terms decay exponentially with rate $\gamma$.

The argument that the quantum dephasing dynamics of a single spin is non-Markovian uses the Shadow-Pocket model [62] as the OQE model for the SE dynamics, for which the reduced dynamics of the system (the spin) can be exactly described by Eq.(25). The Hamiltonian of this OQE model is $H = (g/2)\sigma_z^S \otimes x$ (we set $g = 1$ as the unit), where $x$ is the positional operator of a continuous environment. The SE initial state is set to be a separable state: $\rho_0^{SE} = \rho_0^S \otimes |\psi\rangle\langle\psi|$ with $|\psi\rangle = \sqrt{\gamma/\pi}/(x + i\gamma)$. This model will be referred to as the unitary quantum dephasing model (UQDM) afterwards. The fact that the open quantum dynamics of the spin is non-Markovian can be seen as follows: one prepares an initial state $\rho_0^S$ for the system at time $t_0$ and performs a $\sigma_x^S$ operation on the system at time $t_1$, then the dynamics (for the off-diagonal terms) after $t_1$ will simply be the reverse of that before $t_1$. More generally, if we intervene the quantum dynamics of the system by applying a quantum operation $\Lambda_j$ immediately after time $j\Delta$ for $0 \leq j < k$, then the equality (we denote $\mathcal{E}^S = \exp(\mathcal{L}^{dp}\Delta)$)

$$\rho_{k\Delta}^S = \mathcal{E}^S \Lambda_{k-1} \mathcal{E}^S \Lambda_{k-2} \ldots \mathcal{E}^S \Lambda_0 \rho_0^S, \tag{26}$$

*does not hold* in general. In fact, defining $U = \exp(-ig\Delta x/2)$, one can use Eq.(18) to directly compute the evolution of the effective environment state as

$$\rho_j^E = \frac{1}{2}\left(U\rho_{j-1}^E U^\dagger + U^\dagger \rho_{j-1}^E U\right), \tag{27}$$

with $\rho_0^E = |\psi\rangle\langle\psi|$. As a mixture of two different states, we can see that $\rho_j^E$ will generally be a mixed state for $j > 0$, therefore the memory complexity $\mathcal{C}_j > 0$. And since $\mathcal{N}_j^{osee}$, $\mathcal{N}_j^{ee}$ converges with $\mathcal{C}_j$ for the unitary case, we conclude that the underlying open quantum dynamics is non-Markovian. The memory complexity in this case (thus also $\mathcal{N}^{osee}$ and $\mathcal{N}^{ee}$) can be efficiently computed numerically, which is shown in Fig. 3. We can clearly see that the non-Markovianity increases with $\gamma$, and that it grows much slower than the volume law ($\mathcal{C}_j \propto \log(j)$ is approximately observed from the numerical results).

However, on the other hand, the quantum dephasing dynamics described in Eq.(25) does not have to be generated by the Shadow-Pocket model. It can also be generated, for example, using random unitary operations [63, 64]. Concretely, it can be straightforwardly generated with a quantum Hamiltonian $H = (g/2)\sigma_z^S$ where $g$ subjects to *classical noises* with appropriately chosen noise shapes (Lorentzian). Such a random unitary quantum dephasing model (RUQDM) is also physical and have been implemented in experiment as a way to generate the quantum dephasing channel [65]. In this case, the dynamics of the system will not be reversed by any quantum operations in between, or more generally the equality in Eq.(26) would hold. Therefore the dephasing dynamics generate by the RUQDM should be characterized as Markovian which agrees with the intuition.

The lesson one could learn from this simple example is that even if two open quantum dynamics look perfectly the same when looking at the two-time measurements (quantum map between two times), they could be very different if multi-time quantum measurements are

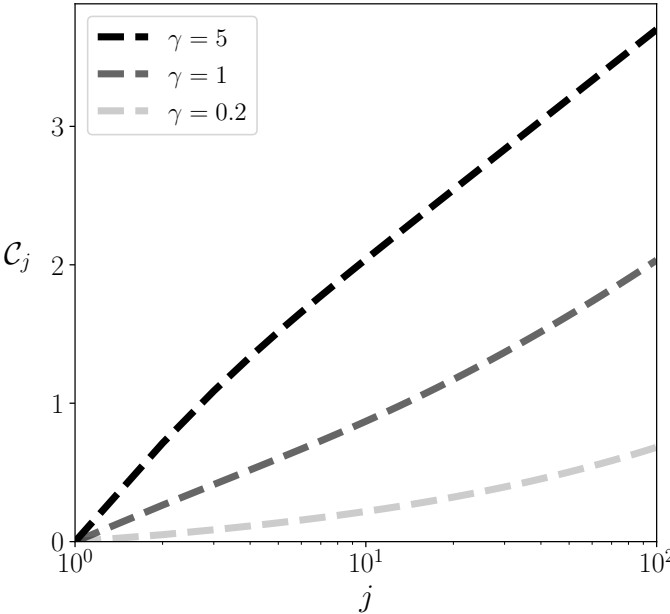

Figure 3: Memory complexity $\mathcal{C}_j$ for the quantum dephasing dynamics of a single spin as a function of the time step $j$, under the open quantum evolution model for the system-environment dynamics. We have used $\Delta = 0.1$, and the continuous environment is discretized into 5000 equidistant points within the interval $[-100\gamma, 100\gamma]$.

considered. Generally, when using the OQE modeling of the open quantum dynamics to study the multi-time correlations, which also involves an infinite number of DOFs, one should not first take the infinite limit and then compute those correlations using Eq.(1), but should directly substitute the OQE into Eq.(1) instead (and then proper infinite limit may be taken). Nevertheless, the process tensor framework gives a way to distinguish those open quantum dynamics which look the same under a quantum map description. In the dephasing case as an example, one could distinguish whether the observed (multi-time) quantum dynamics results from the UQDM or the RUQDM by examining the process tensor (may be obtained from experimental tomography) as follows: one could either compute the memory complexities $\mathcal{C}_j$ and identify the observed dynamics as from the UQDM if $\mathcal{C}_j \approx O(k)$ and from the RUQDM if $\mathcal{C}_j \approx O(\log(k))$, or one could compute $\mathcal{N}_j^{osee}$ or $\mathcal{N}_j^{ee}$ and identify the observed dynamics as from the UQDM if $\mathcal{N}_j^{osee}, \mathcal{N}_j^{ee} > 0$ and from the RUQDM if $\mathcal{N}_j^{osee}, \mathcal{N}_j^{ee} = 0$. In this case, of course, the latter approach could be much more efficient in practice, which also promotes the necessity of this work to define non-Markovianity measures based on the MQE modeling of the SE dynamics.

Additionally, if the observed quantum dynamics indeed follows a Markovian quantum master equation for the system only, then one would have to use an exponentially growing environment to describe it under the OQE model for the SE dynamics, since in general the entanglement entropy of $\Upsilon_{k:0}$ grows linearly with $k$ ($\Upsilon$ is the tensor product of the local $\mathcal{E}^S$s). This fact again shows the necessity of the MQE modeling in complementary to the usual OQE modeling.

It would also be insightful to compare the non-Markovianity measures defined in this work with other existing non-Markovianity measures in literatures, for example, the one proposed in Ref. [4]. In particular, the non-Markovianity measure defined in Ref. [4] vanishes for any divisible quantum dynamics, therefore it is 0 for both UQDM and RUQDM since these two models both satisfy Eq.(25) and their dynamics are divisible. In comparison, our non-Markovianity measures are 0 for RUQDM and nonzero for UQDM (Fig. 3).

# 5 Conclusion

In summary, we have proposed two non-Markovianity measures for general open quantum dynamics, inspired by the Matrix Product Operator and the Matrix Product Density Operator representations of the process tensor respectively. They are fully compatible with the operational Markov criterion proposed in Ref. [16] in the Markovian limit. They can be efficiently calculated given the process tensor in MPO or MPDO form, and are directly related to the complexity of reconstructing the underlying open quantum dynamics. A heuristic algorithm to reconstruct the open quantum dynamics is proposed which reconstructs a hidden MQE model from the observed open quantum dynamics of the system. The non-Markovianity measures as well as the reconstruction algorithm are demonstrated in numerical examples, and the (non-)Markovianity of the quantum dephasing dynamics is carefully reexamined. This work could be very helpful to model and characterize the non-Markovian noises in near-term noisy quantum devices [66, 67].

# Acknowledgments

C. G. acknowledges support from National Natural Science Foundation of China under Grant No. 11805279.

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
