# Peer review of "Quantifying Non-Markovianity in Open Quantum Dynamics"

_SciPost Physics, doi:SciPost Phys. 13, 028 (2022)_

## Round 1 · Referee Report · Anonymous (Referee 1) · 2022-6-14

Strengths

  1. This work provided a method which can be efficiently computed based on the MPO represenation of the process tensor
  2. The computation complexity does not diverge exponentially with the evolution time steps

Report

Quantitatively characterizing the non-Markovianity of a dissipative quantum dynamics is one of the most important topics in the field of open quantum systems. In one of previous work (Ref.[16] in the reference list) , the authors proposed a necessary and sufficient quantum Markov condition based on a process tensor. Such a procedure, even though is sufficiently general, is very difficult, if not impossible, to be implemented in actual calculations, even for simple cases.

The present manuscript by Chu Guo addressed this issue, where he proposed two non-Markovianity measures based on the "entanglement" of the correpsonding MPO representation of the process tensor, which can be used to characterize the "memory effect" of the quantum dynamics. In this measurement, the Markov limit is approached if the process tensor can be written as a separable tensor product state. The application of this method on a concrete example of a two-spin system has also been discuss.

In my opinion, the advantage of the method are two-fold: it can be efficiently computed based on the MPO represenation of the process tensor and the computation complexity does not diverge exponentially with the evolution time steps. For this reason, i think this work is an important advance in the field of open quantum system and recommend the publication of this paper.

---

## Round 1 · Referee Report · Anonymous (Referee 2) · 2022-6-23

Strengths

The paper is clear, complete and exhaustive.
The paper is timely as a lot of work is being dedicated recently to an MPO characterization of non-Markovian time evolution, see e.g. Pollock, PRA 97 012127 (2018); Sonner, Annals of Physics 431 168552 (2021); Guo arXiv:2203:01492 and related works.

Weaknesses

The paper does not offer a direct comparison of the two proposed non-Markovianity measures with other quantities proposed in the literature (such as the one in Breuer, PRL 103 210401 (2008) ).
It is therefore not clear whether the presented criteria are more stringent or more relaxed in measuring non-Markovianity.
It also seems that the complexity is still exponentiall in the spatial size of the system, as each evolution operator of the process tensor is still an operator acting on the Hilbert space of the system; this fact is still crucial when doing numerical simulations and should be stated more clearly.

Report

The paper presents an MPO (and MPDO) representation of the process tensor of the non-Markovian dynamics of a generic open quantum system. This representation is used to define two quantities which quantify the temporal entanglement of the evolution and may be used to characterize the degree of non-markovianity of the system.
The idea is that an MPO naturally describes the degree of entanglement of an operator, akin to what happens for MPS and MPOs in the usual DMRG description of the spatial correlations of a system; in such a sense a state with zero temporal entanglement corresponds to a completely factorizable evolution which corresponds to Markovian dynamics.
The advantage of the proposed non-Markovianity measures is that they scale polinomially in the evolution time instead of exponentially as in other measures.

I recommend publication of the manuscript in SciPost Physics.

---

## Round 2 · Author Response

Reply to Referee 1: I am very grateful for your positive assessment of this work and recommendation to publish on Scipost Physics Reply to Referee 2: I am very grateful for your positive assessment of this work and recommendation to publish on Scipost Physics with some minor comments, which are addressed as follows: 1) "The paper does not offer a direct comparison of the two proposed non-Markovianity measures with other quantities proposed in the literature (such as the one in Breuer, PRL 103 210401 (2008) ). It is therefore not clear whether the presented criteria are more stringent or more relaxed in measuring non-Markovianity" Answer: I thank the referee for this comment. I thus add the last paragraph in the subsection "Reexamining the quantum dephasing dynamics" in the revised manuscript. Basicaly, the Breuer criterion for non-Markovianity will given 0 for both cases considered, namely both models would be considered Markovian due to Breuer criterion. In comparison, with our criterion, one of the two models is Markovian and the other is not. 2) "It also seems that the complexity is still exponentiall in the spatial size of the system, as each evolution operator of the process tensor is still an operator acting on the Hilbert space of the system; this fact is still crucial when doing numerical simulations and should be stated more clearly" Answer: I thank the referee for this comment. Indeed the complexity (of computing the non-Markovianity measures and of the reconstruction algorithm for the hidden MQE) will be exponential with respect to the spatial size of the system in case the system itself is a many-body quantum instead of a single qubit (In the previous manuscript we mainly consider the complexity scaling against the number of time steps). As suggested, I added the second paragraph in the second paragraph in the section "EXAMPLES" to stress the exponential grow of complexity against the spatial dimension of the system in the revised manuscript.

---

## Round 2 · List of Changes

1. The second paragraph in the section "EXAMPLES" is newly added (at the end of the right column on Page 6 and the beginning of the left column on Page 7 )
2. The last paragraph in the subsection "Reexamining the quantum dephasing dynamics" is newly added (at the end of the right column on Page 8)
3. Several words as well as several citations are adjusted (updated) throughout the manuscript.

---

## Editorial Decision

published